# DISTRIBUTION-DRIVEN DISJOINT UNCERTAINTY ESTIMATION FOR DEEP LEARNING

## ABSTRACT

This paper redefines prediction intervals (PIs) as the form of a union of disjoint intervals. PIs represent predictive uncertainty in the regression problem. Since previous PI methods assumed a single continuous PI (one lower and upper bound), it suffers from performance degradation in the uncertainty estimation when the conditional density function has multiple modes. This paper demonstrates that multimodality should be considered in regression uncertainty estimation. To address the issue, we propose a novel method that generates a union of disjoint PIs. Throughout UCI benchmark experiments, our method improves over current state-of-the-art uncertainty quantification methods, reducing an average PI width by over 27%. Through qualitative experiments, we visualized that the multi-mode often exists in real-world datasets and why our method produces high-quality PIs compared to the previous PI.

## 1 INTRODUCTION

Deep neural networks (NNs) show remarkable performance in predicting a target for regression problems. However, the prediction is not enough to make it trustworthy: minimization of objective functions the NN leads to network outputs which approximate the conditional averages of the target data with no information about sampling errors and prediction accuracy. Moreover, if the target is multivalue, NN output can be far from the actual target in the regression problems. Incorporating the predictive uncertainty into the deterministic approximation generated by NNs improves the reliability and credibility of the predictions. This issue is being discussed in various domains such as autonomous driving (Feng et al., 2018), object detection (He et al., 2019), solar energy forecasting (Galván et al., 2017), electricity demands and price estimation (Shrivastava & Panigrahi, 2015), and sensor anomaly detection (Pang et al., 2017).

Prediction interval (PI) represents and quantifies predictive uncertainty in the regression problem. Pearce et al. (2018); Tagasovska & Lopez-Paz (2019); Salem et al. (2020) have recently provided competitive performance by generating a PI to estimate predictive uncertainty. PI describes predictive uncertainty for each sample in the form of two values (lower and upper bound) between which a potential observation falls with a certain probability (e.g., 95% or 99%). PI can provide the amount of uncertainty for each sample by the width of PI. It also provides the possible range of prediction by bounds. It is a self-evident principle that high-quality PI should be as narrow as possible while containing some specified proportion of data points (hereafter referred to as the HQ principle). The quality of a PI is often evaluated by the metric derived from the HQ principle (Khosravi et al., 2010; Galván et al., 2017; Pearce et al., 2018; Tagasovska & Lopez-Paz, 2019; Salem et al., 2020).

Previous methods estimate the regression uncertainty with *a single continuous PI*, but it may suffer from performance degradation in the regression having multimodality. A toy example in Figure 1 is a one-dimensional regression example that has two modes. We observe that *a single continuous PI* (gray shade) provides unnecessarily large PIs to fill in the gap between the two modes compared to disjoint PIs (blue shade). This means that *a single continuous PI* provides low-quality PIs in terms of the HQ principle. Including intervals that are unlikely to contain future observations makes PIs less reliable. Note that this issue becomes severe as the distance between modes increases. We qualitatively confirmed that multimodality often exists in real-world regression datasets through approximating conditional probability density function. We also confirmed that state-of-the-art methods

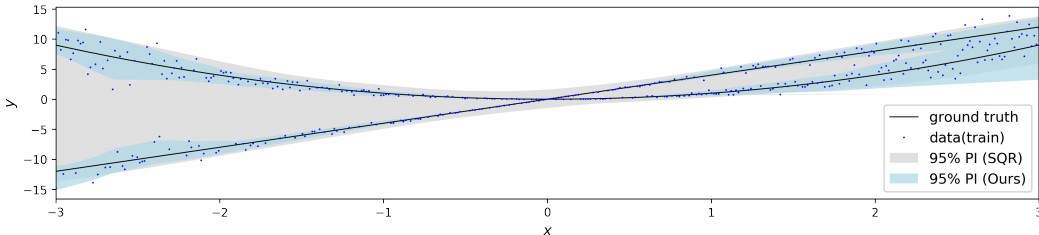

Figure 1: Estimating uncertainty in regression having multimodality. Prediction interval for $y$ with 95% coverage probability is drawn for SQR and DDD (our method). SQR estimates continuous prediction intervals, while DDD provides disjoint prediction intervals.

generate low-quality PIs on real-world samples with multimodality. This is covered in more detail in Section 5.4.

Considering multimodality has been successful at handling the underlying stochastic structure in various fields (Ameijeiras-Alonso et al., 2019; Lerch et al., 2020). Concerning multimodality, various works such as clustering, multi-object detection (Yoo et al., 2019), missing data reconstruction (Smieja et al., 2018), multiple-choice learning (Lee et al., 2017), and multi-output prediction (Guzman-Rivera et al., 2014) have been conducted. However, recent regression uncertainty estimation studies do not consider multimodality in depth.

In this work, we redefine PI as *a union of disjoint PIs* due to the limitation of *a single continuous PI* in multimodality (Section 3). Since prior PI methods and loss functions do not apply to the union of disjoint PIs, we propose a new differentiable objective function and NN architecture that produce the union of disjoint PIs (Section 4). Additionally, we use the ensemble method to boost the performance for both in- and out-of-distribution regions (Section 5.2). As a result, our method improves over current state-of-the-art methods, reducing an average PI width by 27% throughout eleven real-world datasets (Section 5.3). In addition, our method can provide the coverage probability of each disjoint PI (e.g., 20% chance of being between 1 and 3, 75% chance of being between 5 and 9). This means that our method gives information about how reliable each interval is (Section 5.5).

## 2 RELATED WORK

There are two approaches for estimating the predictive uncertainty for regression problems: Bayesian and non-Bayesian. In the Bayesian approach, NN parameters are considered as a distribution, and the uncertainty is calculated by marginalizing the parameters (Graves, 2011; Blundell et al., 2015; Hernández-Lobato & Adams, 2015; Gal et al., 2017; Khan et al., 2018; Wu et al., 2018; Yao et al., 2019; Izmailov et al., 2020). Though theoretically grounded, an approximation is needed since calculating the posterior distribution of NN parameters is computationally intractable. It also requires high computational demand in the inference time. The non-Bayesian approach, on the other hand, defines the output of NN as parameters to describe the predictive uncertainty. It is usually less computational than the Bayesian approach. However, since the NN parameters are fixed, non-Bayesian methods have a limitation in expressing the model uncertainty. Therefore, the deep ensemble with random initialization is additionally used to deal with model uncertainty. Several papers in the non-Bayesian branch have recently provided competitive performance (Lakshminarayanan et al., 2017; Pearce et al., 2018; Tagasovska & Lopez-Paz, 2019; Salem et al., 2020). Our paper focuses on the Non-Bayesian approach, especially for the regression problem. Therefore, we would take a closer look at the non-Bayesian methods by dividing them into PI and non-PI methods.

As PI methods for non-Bayesian methods, Khosravi et al. (2010) propose the Lower Upper Bound Estimation (LUBE) method that produces PI for the first time. Followed by that, Pearce et al. (2018) propose a quality-driven (QD) loss function that is compatible with gradient descent optimization. They also propose an ensemble method for PI with multiple predicted lower and upper bounds to estimate the model uncertainty. Salem et al. (2020) retrofit the QD loss function and propose a new ensemble method by fitting the split normal mixture distribution (Wallis, 2014) to the

PI and averaging the distribution, where they name it as SNM-QD+. It increases the robustness of the training process compared to the QD method. However, SNM-QD+ has difficulties searching hyperparameters because the loss function contains various hyperparameters to achieve the advantages. Tagasovska & Lopez-Paz (2019) propose the simultaneous quantile regression (SQR) and the orthonormal certificates (OC) to estimate data noise and model uncertainty, respectively. However, this strategy generates PI only by the SQR without an ensemble method, and model uncertainty from OC is not included in the PI. Therefore, the PI of SQR does not consider the model uncertainty. Aforementioned loss functions and methods can only generate *a single continuous PI* but not *a union of disjoint PIs*.

As non-PI methods, Mean-Variance Estimation (MVE) (Nix & Weigend, 1994) uses a NN with two output nodes that are considered as a mean and a standard deviation of the conditional probability distribution. Since NN parameters are fixed, it cannot deal with the model uncertainty. Lakshminarayanan et al. (2017) demonstrate the deep ensemble of multiple MVE with random initialization improves the performance, especially in out-of-distribution regions. (so-called $MVE_{ens}$). Fort et al. (2019) shows that ensemble with random initialization may sample different modes in function space and therefore perform well in exploring model uncertainty.

## 3 UNION OF DISJOINT PREDICTION INTERVALS

### 3.1 PROBLEM SETUP

Consider a dataset $\{\boldsymbol{x}_i, y_i\}_{i=1}^N$ where $\boldsymbol{x}_i$ is an input and $y_i$ is a target. For each data point $\{\boldsymbol{x}_i, y_i\}$, the disjoint set of PIs that covers the desired given proportion $\gamma \in [0, 1]$ is defined as follows:

$$PI_i = \bigcup_{j=1}^{J^{(i)}} [L_{ij}, U_{ij}) \tag{1}$$

$$\text{where } Pr(y_i \in PI_i) \geq \gamma \text{ and } L_{ij} \leq U_{ij} < L_{i(j+1)} \text{ for all } j$$

$L_{ij}$ and $U_{ij}$ is a lower and upper bound of $j$th PI related with $i$th data point. $J^{(i)}$ is the number of disjoint intervals when PI is expressed with the smallest number of disjoint intervals. That is, $J^{(i)}$ is unique for a given interval. Note $J^{(i)}$ may have a different value for each data point. The previous methods assume *a single continuous PI* that is $J^{(i)} = 1$ for all $i$.

### 3.2 PERFORMANCE METRIC: PICP AND MPIW

To measure the quality of PI methods based on the HQ principle, let Prediction Interval Coverage Probability ($PICP$) and Mean Prediction Interval Width ($MPIW$) be defined as,

$$PICP = \frac{c}{N} \text{ where } c = \sum_{i=1}^N c_i \text{ and } c_i = \begin{cases} 1, & \text{if } y_i \in PI_i \\ 0, & \text{otherwise} \end{cases} \tag{2}$$

$$MPIW = \frac{1}{N} \sum_{i=1}^N \sum_{j=1}^{J^{(i)}} (U_{ij} - L_{ij}) \tag{3}$$

$PICP$ measures the ratio of the target that is captured within PIs while $MPIW$ measures total length of PIs over the entire samples. According to the HQ principle, PIs should minimize $MPIW$ subject to $PICP \geq \gamma$ (e.g. $\gamma = 0.95$ or $0.99$). This metric is widely used to compare performance of PI-related methods (Khosravi et al., 2010; Pearce et al., 2018; Tagasovska & Lopez-Paz, 2019; Salem et al., 2020).

## 4 DISTRIBUTION-DRIVEN-DISJOINT METHOD

We propose a learning-based method that generates distribution-driven disjoint (DDD) PIs, and we call our method as DDD method. The DDD method produces high-quality PIs without the assumption that $J^{(i)} = 1$. To generate multiple disjoint PIs with a learning-based method, we need to

formulate a differentiable loss function that reflects the HQ principle. However, this is a challenging problem because $c$ from (2) is non-differentiable. Pearce et al. (2018); Salem et al. (2020) proposed a QD and QD+ loss function in the form of constraint optimization by approximating $c$ in a differentiable way. Tagasovska & Lopez-Paz (2019) employed a pinball loss which reflects HQ principle and differentiable. However, these loss functions have a limitation in that they work for a single continuous PI which has $J^{(i)} = 1$.

To derive a new differentiable loss function (so-called DDD loss) for multiple disjoint intervals, we first approximate the conditional distribution given input, $\hat{p}(y_i|x_i)$, with a Gaussian mixture model. Then, we derive the DDD loss by using the cumulative density function of the Gaussian mixture (why we call our method distribution-driven disjoint). Our DDD method trains NN by minimizing the DDD loss.

Another major problem is that the optimal number of disjoint prediction intervals $J_{opt}^{(i)}$ may differ for each $\hat{p}(y_i|x_i)$, making it hard to implement $[\hat{L}_{ij}, \hat{U}_{ij}]_{j=1}^{J^{(i)}}$ as an output of NN. To deal with the problem, after producing $K$ intervals regardless of conditional density, the union process removes the overlapping part. We propose a novel architecture that can implement this in a differentiable way. Additionally, we employed a simple ensemble method to improve performance for both in- and out-of-distribution observations.

## 4.1 DERIVATION OF THE DDD LOSS

In this section we derive the DDD loss, $L_{DDD}$, that reflects the HQ principle: PIs should minimize $MPIW$ subject to $PICP \geq \gamma$. $L_{DDD}$ combine $L_{MPIW}$ and $L_{PICP}$ with a Lagrangian, $\lambda$, controlling the importance of width vs. coverage. Each term is defined as:

$$L_{DDD} = L_{MPIW} + \lambda L_{PICP} \tag{4}$$

$$L_{PICP} = \max(0, \gamma - \frac{1}{N}\sum_{i=1}^{N}\hat{c}_i)^2, \quad L_{MPIW} = \frac{1}{N}\sum_{i=1}^{N}\sum_{j=1}^{J^{(i)}}(\hat{U}_{ij} - \hat{L}_{ij}) \tag{5}$$

$$\text{where } \hat{c}_i = \sum_{j=1}^{J^{(i)}}(F_i(\hat{U}_{ij}) - F_i(\hat{L}_{ij})). \tag{6}$$

The coverage probability $\hat{c}_i$ is an approximation of $c_i$ in Equation (2). The function $F_i$ is a cumulative density function of the distribution $\hat{p}(y_i|x_i)$. Then, the coverage probability $\hat{c}_i$ is calculated from the cumulative density function $F_i$. $L_{PICP}$ is defined as the mean square error with the target proportion $\gamma$ with the $max(0, x)$ operation. The $max(0, x)$ operation is used to minimize the loss only when the average coverage probability, $\frac{1}{N}\sum_i \hat{c}_i$, is lower than the given proportion $\gamma$. The loss function $L_{MPIW}$ is naively derived from the definition of $MPIW$ in equation (3).

For $L_{DDD}$ to be differentiable for $\hat{L}_{ij}$ and $\hat{U}_{ij}$, $\hat{c}_i$ must be differentiable for $\hat{L}_{ij}$ and $\hat{U}_{ij}$. Therefore, $F_i$ should be differentiable. We approximate the $p(y_i|x_i)$ with the mixture density network with Expecation-Maxmization algorithm (Greff et al., 2017). This is not a distributional assumption because any distribution can be approximated by the Gaussian mixture model when the number of the mixture is sufficient (Bishop, 1994; Sung, 2004). When parameters of the Gaussian mixture distribution are $\{\boldsymbol{\mu}_i, \boldsymbol{\sigma}_i, \boldsymbol{\pi}_i\}$, an approximated probability distribution $\hat{p}(y_i|\boldsymbol{x}_i)$ is defined as

$$f(\boldsymbol{x}_i; \boldsymbol{\mu}, \boldsymbol{\sigma}, \boldsymbol{\pi}) = \sum_{k=1}^{K}\pi_k(\boldsymbol{x}_i)\,\mathcal{N}(y_i \,|\, \mu_k(\boldsymbol{x}_i), \sigma_k^2(\boldsymbol{x}_i)), \tag{7}$$

where $K$ is the number of mixture components. $K$ can be considered large enough if increasing $K$ does not reduce the negative likelihood loss of the mixture density network. Setting K=5 worked well for our experiment in section 5. For brevity, we refer to $\mu_k(\boldsymbol{x}_i)$ as $\mu_{ik}$, and same abbreviation applied to $\sigma_k$ and $\pi_k$.

---

**Algorithm 1** Pseudocode for the DDD Network

---

1: **Definition**
2:     $\boldsymbol{\mu}_i, \boldsymbol{\sigma}_i, \boldsymbol{\pi}_i$ : Input of NN where $\texttt{size}(\boldsymbol{\mu}_i) = \texttt{size}(\boldsymbol{\sigma}_i) = \texttt{size}(\boldsymbol{\pi}_i) = K$.
                The means, standard deviations, and weights of the Gaussian mixture $\hat{p}(y_i|\boldsymbol{x}_i)$.
3:     $\boldsymbol{\epsilon}_i^{(-)}, \boldsymbol{\epsilon}_i^{(+)}$ : Output of NN where $\texttt{size}(\boldsymbol{\epsilon}_i^{(-)}) = \texttt{size}(\boldsymbol{\epsilon}_i^{(+)}) = K, \boldsymbol{\epsilon}_i^{(-)} \geq 0, \boldsymbol{\epsilon}_i^{(+)} \geq 0$

4: **Train**
5:     $\hat{\boldsymbol{L}} = \boldsymbol{\mu} - \boldsymbol{\epsilon}^{(-)}$ and $\hat{\boldsymbol{U}} = \boldsymbol{\mu} + \boldsymbol{\epsilon}^{(+)}$.
6:     $\hat{\boldsymbol{L}} = \hat{\boldsymbol{L}}[\texttt{argsort}(\hat{\boldsymbol{L}})], \hat{\boldsymbol{U}} = \hat{\boldsymbol{U}}[\texttt{argsort}(\hat{\boldsymbol{L}})]$          ▷ ascending order in $k$-related dimension
7:     **for** $i \leftarrow 1 \cdots N$ **do**
8:         **for** $k \leftarrow 1 \cdots K$ **do**
9:             $\hat{U}_{ik} = \hat{L}_{ik} + \texttt{ReLU}(\hat{U}_{ik} - \hat{L}_{ik})$
10:            $\hat{L}_{i(k+1)} = \hat{U}_{ik} + \texttt{ReLU}(\hat{L}_{i(k+1)} - \hat{U}_{ik})$ **if** $k \neq K$
11:        **end for**
12:    **end for**
13:    Minimize $L_{DDD} = \frac{1}{N} \sum_{i=1}^{N} \sum_{j=1}^{K} (\hat{U}_{ik} - \hat{L}_{ik}) + \lambda \max(0, \gamma - \frac{1}{N} \sum_{i=1}^{N} \hat{c}_i)^2$
                where $\hat{c}_i = \sum_{j=1}^{K} (F_i(\hat{U}_{ik}) - F_i(\hat{L}_{ik}))$

---

## 4.2 DDD NETWORK ARCHTECTURE

This section describes a neural network architecture that generates distribution-driven-disjoint PIs (so-called DDD network) as pseudocode in Algorithm 1. The architecture is classified into four blocks according to the major objectives and explained in detail. All blocks achieve each objective in a differentiable form.

**Input Generation (line 2)**   Input of the DDD network is $[\boldsymbol{\mu}_i, \boldsymbol{\sigma}_i, \boldsymbol{\pi}_i]$ because $\boldsymbol{\mu}_i$, $\boldsymbol{\sigma}_i$, and $\boldsymbol{\pi}_i$ are used to compute the loss function. The input $\boldsymbol{x}_i$ is converted into $[\boldsymbol{\mu}_i, \boldsymbol{\sigma}_i, \boldsymbol{\pi}_i]$ by the mixture density network. For this, it is necessary to train the mixture density network before training the DDD network. The Neural Expectation-Maximization algorithm (Greff et al., 2017) is used for the training.

**Multiple Interval Generation (line 3-5)**   This block generates $K$ multiple intervals. The last nodes of NN are squared to ensure that the outputs of NN ($[\boldsymbol{\epsilon}_i^{(-)}, \boldsymbol{\epsilon}_i^{(+)}]$) are positive. The primary aspect is that this block restricts each interval to include each peak $\mu_{ik}$ of the Gaussian mixture model (line 5). This aspect contributes to producing well-calibrated PIs because it is self-evident that well-calibrated PIs should contain more than one peak. If the output of NN is defined as lower bounds and upper bounds without the restriction, it is experimentally confirmed that it often converges to low-quality PI.

**Union: overlap prevention (line 6-12)**   This block implements the union process in a differentiable way using sorting and $max(0, x)$ functions (the sorting process does not interfere with back-propagation). Through this process, $K$ intervals can be converted to $J^{(i)}$ blocks. As a result, the DDD method provides the union of disjoint PIs.

**Minimize $L_{DDD}$ (line 13).**   By training the DDD network to minimize $L_{DDD}$, the union of disjoint PIs becomes high-quality. $\lambda$ is a hyperparameter that determines how much trade-off between $L_{PICP}$ and $L_{MPIW}$. If $PICP$ obtained through training is less than $\gamma$, increase $\lambda$. This increments $PICP$, along with $MPIW$.

## 4.3 ENSEMBLE METHOD

Optimizing the several NN models with random initialization and then aggregating them is known to be effective for the uncertainty estimation in the out-of-distribution region (Lakshminarayanan et al., 2017). Since then, recent studies (Pearce et al., 2018; Salem et al., 2020) have promoted the ensemble of PIs.

We train $M$ different mixture density newtork models for $\hat{p}(y_i|\boldsymbol{x}_i)$ with random initialization followed by averaging them, $f^{ens}(x_i) = \frac{1}{M} \sum_m f^{(m)}(x_i)$, where $f^{(m)}(\cdot)$ is the probability density

function in equation (7). Then, we can acheive cumulative density function of the ensembled model from $F^{ens}(x_i) = \frac{1}{M} \sum_m F^{(m)}(x_i)$. Finally, we train the DDD network proposed in 4.2 to achieve estimating the predictive uncertainty. When $M$ models are ensembled, not $K$ but $KM$ intervals are generated in the multiple interval generation block.

## 5 EXPERIMENT

### 5.1 IMPLEMENTATION DETAIL

**Dataset**   We use UCI regression benchmarking datasets (Dua & Graff, 2017).  While previous works use nine datasets (Boston, Energy, Kin8nm, Naval, Power, Protein, Wine, and Yacht) for evaluation, we add two more datasets that are the Parkinson-Telemonitoring and Bike-Sharing. For each dataset, the experiment is repeated 20 times with different random seeds except for the Protein dataset, where it is repeated five times. Each dataset is split into train, valid, and test set with the ratio of 0.81, 0.09, and 0.1, with random shuffling for each experiment. The target $y$ is standardized to zero mean and unit variance based on the entire data in each dataset. More specifics about data pre-processing are provided in the Appendix A.

**Methods**   We compare DDD (ours) with MVE (Lakshminarayanan et al., 2017) and QD+ (Salem et al., 2020) where the ensemble method of each of them are named $DDD_{ens}$, $MVE_{ens}$, and SNM-QD+, respectively.  SQR (Tagasovska & Lopez-Paz, 2019) participates in the comparison without an ensemble since Tagasovska & Lopez-Paz (2019) did not propose an ensemble method for its PI generation. Considering QD+ (Salem et al., 2020) is a retrofit version of QD (Pearce et al., 2018), QD is not included in the comparison. We build the PI of the $MVE_{ens}$ method based on its mean and standard deviation like Pearce et al. (2018) did.

**Hyperparameters**   While a larger $K$ (the number of mixtures) is preferable in theory (Sung, 2004), it needs more data and increased model complexity in practice. In our method, $K = 5$ was sufficient to apply in the UCI datasets. We experimented with the same hyperparameters for all methods as follows: All neural networks have two hidden layers with 50 units except for the Protein dataset, where 100 units are used, with each layer having the rectified linear activation function. We set $\gamma = 0.95$ for the ratio of PIs to be covered. The ensemble size is set to $M = 5$. Adam optimizer (Kingma & Ba, 2014) is used for the optimization, and we did not use the dropout (Srivastava et al., 2014) and the weight decay.

For a fair comparison, hyperparameters should be set to derive near-optimal performance for each method. The grid search is used for the hyperparameter search on the learning rate for $MVE_{ens}$ and the mixture density network of the $DDD_{ens}$ method. For the $DDD_{ens}$ network, we search only on the lambda contained in the loss function (the same learning rate was used as the first phase). However, in the case of SNM-QD+, it is difficult to optimize because the loss function includes multiple hyper-parameters. Therefore, we take the quantitative results from Salem et al. (2020) for the nine existing UCI datasets. For the additional two datasets, we run 300 hyperparameter searches with the code provided by the paper's authors.

We set $\lambda$ so that $DDD_{ens}$ has $PICP$ similar to other methods. It is easy to compare the performance between methods from the perspective of the HQ principle if the $PICP$ is similar. This is because if $PICP$ is similar, only $MPIW$ needs to be compared. Note that, as $\lambda$ increases, PICP and MPIW increase, and vice versa.

### 5.2 EFFECT OF THE ENSEMBLE METHOD

Figure 2 presents qualitative results to confirm the effects of ensemble in in-distribution and out-of-distribution. It describes uncertainty estimation of three single DDD models and the ensemble of the three. The blue dotted line means the amount of uncertainty (yticklabels on the right side). Data points are generated through $y = \epsilon_1(0.02x^3 + 0.02\epsilon_2) + (1 - \epsilon_1)(x + 0.02\epsilon_2)$ where $\epsilon_1 \sim Bernoulli(\frac{1}{2})$ and $\epsilon_2 \sim N(0, 3^2)$. Training data points are not included in $x < -4$ and $x > 4$ to demonstrate the effect of $DDD_{ens}$ at those out-of-distribution regions. The models used in Figure 2 are the result of adjusting the $\lambda$ value so that $PICP$ for the validation set is $0.95 \pm 0.01$. Since the data points are generated to have homoscedastic noise, the amount of uncertainty in in-distribution

Figure 2: Qualitative comparison of DDD and DDD$_{ens}$. DDD is trained with the training dataset (red dots). Gray shade and blue shade represent the estimated prediction intervals of DDD and DDD$_{ens}$, respectively. The Blue dotted line represents the width of prediction intervals for each input. DDD$_{ens}$ shows better calibration in both in-distribution and out-of-distribution.

is almost constant. In out-of-distribution, the amount of uncertainty increase toward both ends. In other words, the blue dotted line should have a U shape if high-quality. We can observe that the ensemble method produces more tight PIs for the in-distribution region (compared to the left side of #1) and quantifies better the uncertainty for the out-of-distribution region (compared to both sides of #1 and the left side of #2 and #3). Put simply, the ensemble model has a clearer U shape compared to other single models.

We quantitatively compared DDD$_{ens}$ with a single DDD model on the UCI benchmark datasets. According to the HQ principle, when $PICP$ is greater than or equal to, and $MPIW$ is smaller, better performance is achieved. For all datasets, we confirmed that DDD$_{ens}$ achieves higher $PICP$ than DDD while having lower to equal $MPIW$ that implies outperformance of DDD$_{ens}$. The comparison result is presented in appendix B.

Combining all the experimental results, we can conclude that our ensemble method boosts the performance for both in- and out-of-distribution.

## 5.3 BENCHMARKING EXPERIMENT

Table 1 compares DDD$_{ens}$, MVE$_{ens}$, SNM-QD+ and SQR on the 11 UCI datasets. For each datasets, we compare DDD$_{ens}$ with the best performance method among MVE$_{ens}$, SNM-QD+ and SQR and measure how much MPIW is reduced. Improvement means how much DDD$_{ens}$ reduces MPIW. DDD$_{ens}$ and the best method for each dataset are marked in bold. It is a better method if the method has a higher or equal $PICP$ and a smaller $MPIW$. However, if both $PICP$ and $MPIW$ are higher or smaller at the same time, comparison is not clear. Therefore, we set the criterion that the best performance method of each dataset is determined to have the smallest $MPIW$ among those with a $PICP$ of 0.94 or higher.

DDD$_{ens}$ has higher or equal $PICP$ and a smaller $MPIW$ compared to the best performance methods in 10 out of 11 datasets. This confirms that DDD$_{ens}$ provides well-calibrated PIs for predictive uncertainty estimation compared to the state-of-the-art methods. We can observe minor improvements like 1% or 3%, and dramatic improvements like 72% or 93%. It is reasonable to infer that datasets with significant improvement have multimodality, and those with low improvement have unimodality. The reason why DDD$_{ens}$ shows better performance and multimodality of the datasets are discussed in Section 5.4 through a qualitative experiment.

## 5.4 QUALITATIVE EXPERIMENT

We draw $\hat{p}(y_i|\boldsymbol{x}_i)$ through the mixture density network for the following reasons: 1. to find out whether the conditional probability density of samples is multimodal. 2. to ascertain the reason for the outperformance of DDD$_{ens}$. Each plot in Figure 5 is the sample of the Protein, Parkinson, Kin8, and Energy dataset. The light blue line denotes the approximated the conditional density $\hat{p}(y_i|\boldsymbol{x}_i)$. Note that, although approximated by a Gaussian mixture model, $\hat{p}(y_i|\boldsymbol{x}_i)$ can express a unimodal distribution (See Kin8 samples and Power-Plant sample #1).

In the Protein and Parkinson samples, we observe that our mixture model describes a multimodal distribution. DDD provides multiple disjoint prediction intervals covering the target and does not contain less plausible regions than SQR and MVE. In other words, even with a much shorter PI,

Table 1: Quantitative comparison among three different methods.

| | Method | bike | boston | concrete | energy | kin8 | naval |
|---|---|---|---|---|---|---|---|
| PICP | SQR | 0.92±0.01 | 0.84±0.06 | 0.88±0.06 | 0.84±0.04 | 0.92±0.01 | 0.95±0.02 |
| | $MVE_{ens}$ | **0.96±0.01** | **0.95±0.02** | 0.95±0.03 | **0.97±0.02** | 0.98±0.01 | **1.00±0.01** |
| | SNM-QD+ | 0.96±0.00 | 0.95±0.01 | **0.94±0.01** | 0.99±0.00 | **0.97±0.00** | 1.00±0.00 |
| | $DDD_{ens}$ | **0.96±0.01** | **0.95±0.03** | **0.94±0.03** | 0.98±0.02 | 0.97±0.01 | 1.00±0.00 |
| MPIW | SQR | 1.38±0.10 | 0.92±0.26 | 0.92±0.16 | 0.11±0.02 | 0.98±0.05 | 0.14±0.02 |
| | $MVE_{ens}$ | **1.62±0.07** | **1.43±0.19** | 1.08±0.18 | **0.14±0.01** | 1.11±0.02 | **0.09±0.01** |
| | SNM-QD+ | 1.82±0.1 | 1.58±0.06 | **0.99±0.04** | 0.29±0.01 | **1.07±0.01** | 0.09±0.00 |
| | $DDD_{ens}$ | **1.20±0.04** | **1.21±0.12** | **0.96±0.07** | 0.16±0.01 | 1.03±0.02 | 0.06±0.01 |
| | IMPROVEMENT | 26% | 15% | 3% | NA | 4% | 33% |
| | Method | parkinson | power | protein | wine | yacht | - |
| PICP | SQR | 0.90±0.02 | **0.94±0.01** | 0.94±0.00 | 0.91±0.04 | 0.82±0.09 | - |
| | $MVE_{ens}$ | **0.99±0.00** | 0.96±0.01 | 0.97±0.00 | 0.95±0.02 | 0.99±0.02 | - |
| | SNM-QD+ | 0.97±0.01 | 0.95±0.00 | **0.95±0.00** | **0.94±0.01** | **0.94±0.01** | - |
| | $DDD_{ens}$ | **0.99±0.01** | **0.96±0.01** | **0.95±0.00** | 0.94±0.02 | 0.94±0.04 | - |
| MPIW | SQR | 1.79±0.13 | **0.79±0.04** | 2.25±0.05 | 2.51±0.51 | 0.08±0.03 | - |
| | $MVE_{ens}$ | **1.43±0.10** | 0.84±0.03 | 2.46±0.07 | 3.33±0.47 | 0.15±0.04 | - |
| | SNM-QD+ | 1.68±0.10 | 0.80±0.00 | **2.12±0.01** | **2.62±0.06** | **0.12±0.00** | - |
| | $DDD_{ens}$ | **0.40±0.05** | **0.79±0.02** | **1.57±0.02** | **0.18±0.05** | **0.09±0.01** | - |
| | IMPROVEMENT | 72% | 1% | 26% | 93% | 25% | - |

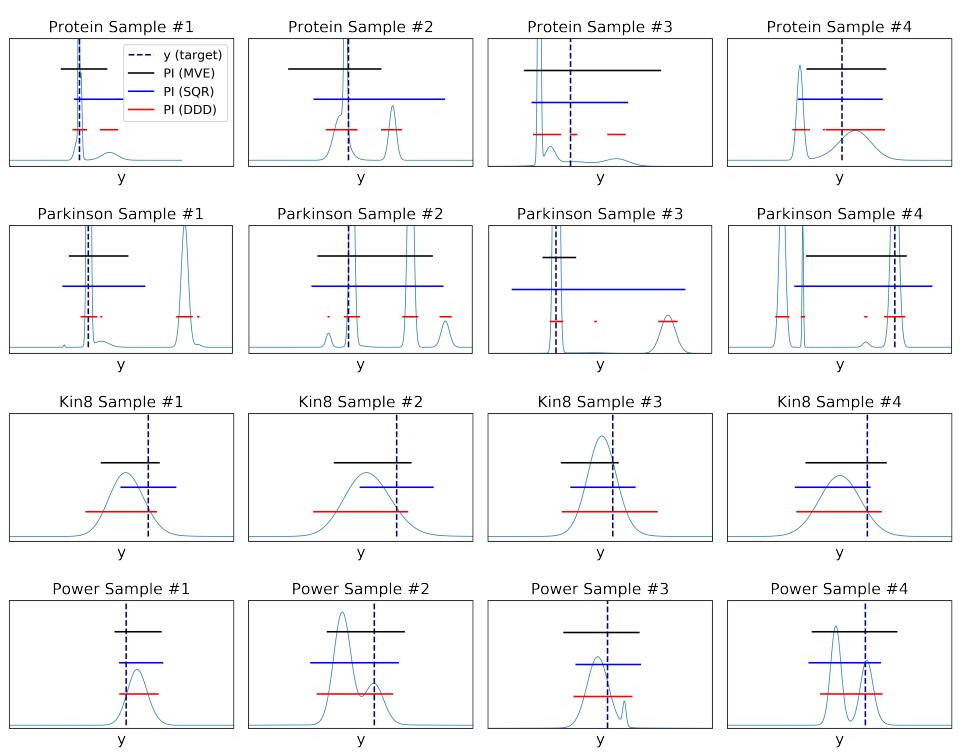

Figure 3: Qualitative comparison between DDD, SQR, and MVE. Blue dotted line represents target value. The solid red lines, blue lines, and black lines are the PIs of DDD, SQR, and MVE repectively. DDD, SQR, and MVE were configured to have the same $PICP$ for a fair comparision.

DDD can cover future outcomes. This explains the outperformance of $DDD_{ens}$ having 26% and 72% improvement. In the Kin8 samples, $\hat{p}(y_i|\boldsymbol{x}_i)$ is very close to unimodal. In the Power sample #2, #3, and #4, $\hat{p}(y_i|\boldsymbol{x}_i)$ are bimodal, but the distance between modes is short so that DDD has a

union of PIs that looks like a single continuous PI. The length of each PI generated by the three methods does not differ significantly in Kin8 and Power samples. This explains the reason why there is a small improvement in Kin8 and Power datasets.

There may be some misspecification because $\hat{p}(y_i|\boldsymbol{x}_i)$ approximated by the mixture density network is not ground-truth. Even considering the misspecification, the experimental results are sufficient to explain that multimodality exists in the real world data and the reason for the outperformance of our method.

## 5.5 PIS WITH COVERAGE PROBABILITY

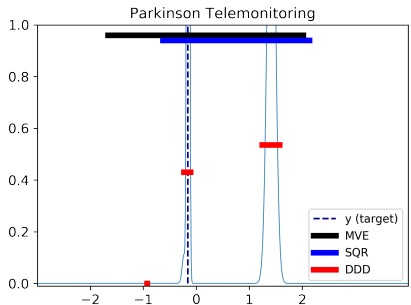

Figure 4: PIs (DDD) with coverage probability. Left scale means probability.

This section covers additional advantages of our method. One advantage of DDD is that it contains coverage probability for each PI, giving critical information about how reliable each Pi is. Figure 4 represents the uncertainty estimation in one sample from the Parkinson dataset. The y-axis of plots denotes the coverage probability for each PI. Our method provides three PIs (red line) with close to 0%, 43%, and 54% probability, respectively. We can neglect the leftmost PI and rely more on the other two PIs. Another advantage is that we can presume the number of the mode of each sample. The information that two distinct PIs are not negligible implies that the sample has two modes. On the other hand, the previous single-PI methods do not provide such information.

## 6 CONCLUSION

This paper proposes a novel method that generates disjoint PIs and demonstrates the need for disjoint PIs to estimate the predictive uncertainty of the regression problems. We confirm that the ensemble method improves performance for estimating uncertainty in both in- and out-of distribution. Through the UCI benchmarking datasets, we have shown quantitatively that the DDD method outperforms state-of-the-art methods. The DDD method improves over current state-of-the-art methods, reducing an average PI width by over 27%. Additionally, by approximating the conditional density function, we showed that multimodality often exists in real-world data, and we qualitatively explained why DDD generates well-calibrated PIs compared to previous methods. Our method also provides coverage probability corresponding to each PI that is simple but informative summarization of the conditional probability of the target. We can conclude that our method can provide well-calibrated and informative PIs for predictive uncertainty estimation in various fields.

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

## SUPPLEMENTARY MATERIALS

## A   DATA PRE-PROCESSING

We use the same dataset and pre-processing implemented in SNM-QD+ Salem et al. (2020) [1]. We add two more datasets that are the Parkinson-Telemonitoring and Bike-Sharing from UCI datasets:

Parkinson Telemonitoring[2]: "subject id" is excluded
Seoul Bike Sharing Demand [3]: "date", "seasons", "holiday", and "functioning day" are excluded.

## B   QUANTITATIVE COMPARISON BETWEEN DDD AND $\text{DDD}_{ens}$

Table 2 compares $\text{DDD}_{ens}$ with a single DDD model on the UCI benchmark datasets. $\text{DDD}_{ens}$ has higher $PICP$ and lower or equal $MPIW$ over DDD throughout all datasets except for the Yacht. In the Yacht dataset, $\text{DDD}_{ens}$ has similar $MPIW$ and much higher $PICP$, so it can be considered that $\text{DDD}_{ens}$ is high-quality. This result implies that the ensemble method improves performance in terms of the HQ principle.

Table 2: Quantitative comparison between DDD and $\text{DDD}_{ens}$.

|  | Method | bike | boston | concrete | energy | kin8 | naval |
|---|---|---|---|---|---|---|---|
| PICP | DDD | 0.94±0.01 | 0.92±0.03 | 0.90±0.05 | 0,89±0.04 | 0.94±0.01 | 0.97±0.00 |
|  | $\text{DDD}_{ens}$ | **0.96±0.01** | **0.95±0.03** | **0.94±0.03** | **0.98±0.02** | **0.97±0.01** | **1.00±0.00** |
| MPIW | DDD | 1.23±0.07 | 1.30±0.26 | 1.00±0.18 | 0.12±0.02 | 1.07±0.05 | 0.06±0.02 |
|  | $\text{DDD}_{ens}$ | **1.20±0.04** | **1.21±0.12** | **0.96±0.07** | **0.16±0.01** | **1.03±0.02** | **0.06±0.01** |

|  | Method | parkinson | power | protein | wine | yacht | - |
|---|---|---|---|---|---|---|---|
| PICP | DDD | 0.95±0.01 | 0.95±0.01 | 0.94±0.00 | 0.94±0.02 | 0.86±0.07 | - |
|  | $\text{DDD}_{ens}$ | **0.99±0.01** | **0.96±0.01** | **0.95±0.00** | **0.94±0.02** | 0.94±0.04 | - |
| MPIW | DDD | 0.43±0.13 | 0.80±0.02 | 1.64±0.04 | 0.26±0.22 | 0.08±0.02 | - |
|  | $\text{DDD}_{ens}$ | **0.40±0.05** | **0.79±0.02** | **1.57±0.02** | **0.18±0.05** | 0.09±0.01 | - |

---

[1] The code from the author is available at `https://github.com/tarik/pi-snm-qde`.
[2] `https://archive.ics.uci.edu/ml/datasets/parkinsons+telemonitoring`
[3] `https://archive.ics.uci.edu/ml/datasets/Seoul+Bike+Sharing+Demand`

# C   MORE MULTIMODAL SAMPLES

We draw $\hat{p}(y_i|\boldsymbol{x}_i)$ through the mixture density network in the same way as Section 5.4. The Bike Sharing dataset shows clear multimodality and DDD produce high-quality PIs compared to MVE and SQR. The Wine dataset has a discretized outcome because it is for the regression problem of guessing the grade of wine. DDD generates very short intervals for each grade position compared to others. For datasets with discretized outcomes like the Wine samples, it is much more helpful to obtain the coverage probability for each PI than the total width of PIs (e.g., grade 1 with a probability of 20%, grade 2 with a probability of 45%, and grade 3 with a probability of 30%).

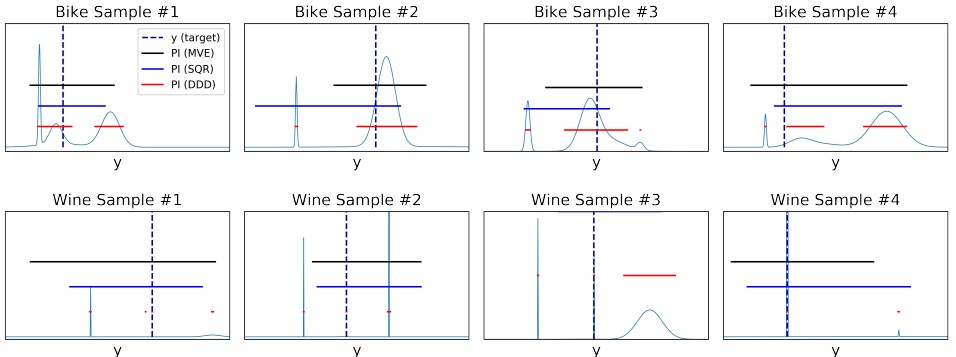

Figure 5: Qualitative comparison between DDD, SQR, and MVE. Blue dotted line represents target value. The solid red lines, blue lines, and black lines are the PIs of DDD, SQR, and MVE repectively.

