# OpenReview forum: "Distribution-Driven Disjoint Prediction Intervals for Deep Learning"
_ICLR.cc/2022/Conference — ICLR 2022 Submitted_

### Official Review · Reviewer_3Zfh · 2021-10-31

**Correctness:** 2
**Technical Novelty And Significance:** 3
**Empirical Novelty And Significance:** 2
**Recommendation:** 3
**Confidence:** 3

**Main Review:**

This paper makes a strong and somewhat overlooked point that existing methods for the evaluation of PIs provide low fidelity in the presence of multimodality. The recent introduction of flexible methods for heteroscedastic noise modelling, such as SQR, creates a clear gap in the literature regarding their evaluation. The paper under consideration here attempts to fill this gap. However, from my point of view, it has 2 critical shortcomings that prevent it from representing a significant contribution.

1. The two key contributions of the paper are difficult to disentangle. As a result, I do not think the experiments validate the main claims of the paper.
	* According to the authors, existing methods provide single predictive intervals and thus can not be evaluated using the tighter multi-interval method provided in equation 1. I do not agree with this point. It is not clear to me why a potentially multimodal predictive distribution, like SQR, can not generate disjoint PIs. A simple procedure to do this would be to place the centre of these PIs at the top-K modes of the predictive distribution. This is not very different from what the authors already do with their GMM, which also has full support in $\mathcal{R}$.
	* As a result of the above issue, the authors evaluate their method using a disjoint set of PIs while evaluating baselines using a single PI. It seems clear that the PICP / MPIW tradeoff obtained by the multimodal generation+evaluation approach produces better results. However, there is no evidence that the proposed PI generation method is any better than competing approaches, which could potentially also produce multimodal distributions. For instance, in figure 3 and figure 5, SQR seems to always cover the right range in output space but its PI is always spread too thin due to multimodality.


2. The motivation for the proposed PI generation method is unclear to me.
	* The authors propose to use a mixture density network combined with an auxiliary NN that outputs PI lower and upper bounds from the GMM parameters. It is not clear to me why they need the auxiliary network considering that disjoint $\gamma$ % PIs can be obtained in closed form from the mixture density network directly. Is this last step intended to relax the Gaussianity assumption of the mixture density network?

I am willing to raise my score to an accept if the authors clarify the motivation for their PI generation approach and evaluate competing multimodal noise modelling approaches using disjoint PIs.

**Other comments:**

The paper has many grammar errors. I would recommend the use of an online grammar checker.


**Summary Of The Paper:**

This paper notes that existing approaches to predictive interval (PI) generation and evaluation assume unimodal predictive distributions. This results in unnecessarily loose PIs in the presence of multimodality in the predictive distribution or multimodality in the targets. The authors first propose to extend the definition of PI to be a union of disjoint intervals, allowing for more fidelity in evaluation. They then propose a method to explicitly generate multimodal PIs. This method is based on feeding the output of a mixture density network (conditional GMM) through a secondary NN that outputs a set of lower and upper bounds for PIs. The authors compare their method against alternative heteroscedastic noise models in terms of PICP and MPIW on 11 UCI datasets.

**Summary Of The Review:**

This paper proposes a higher fidelity method for the evaluation of predictive intervals under multimodal predictions. The authors also propose a method to make multimodal predictions. Unfortunately, the author’s evaluation setup assumes that their proposed method is the only one to produce multimodal predictions (which is not the case). This creates a drastically unfair comparison.

---

> ### Author Response · Authors · 2021-11-15
> **Response to 3Zfh**
>
> Thank you for your interest and insightful feedback on our research.
>
> You have raised the point that we tried early in the study to generate $\gamma$ % disjoint PI.  We tried a slight modification of the existing method or simple ideas but those did not produce high-quality PIs. (Note that high-quality PI should be as narrow as possible while containing some specified proportion ($\gamma$) of data points.). Therefore, we understand why you gave such feedback and can explain it below clearly.
>
> [Response to the feedback]
>
> According to the authors, existing methods provide —
>
> - First, since the predictive distribution is unknown, it should be approximated by a method such as a mixture density network.
> - According to your feedback, given approximated predictive distribution, we need to find the PIs (centered on each mode) which cover $\gamma_1$%, $\gamma_2$%, ... $\gamma_K$% respectively. PIs should satisfy $\gamma_1+\gamma_2+...\gamma_K=\gamma$, however, we have no information about $\gamma_1, \gamma_2,..., \gamma_K$ so that it is not possible to generate $\gamma$ % PIs around each mode.
> - The previous loss functions were driven to generate only one pair of lower bound and upper bound. We have tried to extend the loss function to create multiple lower bounds and upper bounds, but it was a completely different driven process and challenging. To the best of our knowledge, a method for generating multiple PIs has not yet been proposed.
> - Instead, our method generates PIs from the constrained optimization problem: minimize the width of PIs while satisfying $\gamma_1+\gamma_2+...+\gamma_K=\gamma$.
>
> As a result of the above issue, the authors evaluate —
>
> - As far as we know, our method is the only method to solve the multimodality issue by generating multiple PIs in the field of predictive uncertainty estimation. Generating well-calibrated multiple PIs is a seemingly simple but challenging problem.
> - Even if there are other methods, the fact that we are the first to point out the limitation that the state-of-the-art methods do not sufficiently consider multimodality and contribute to further research in the field of predictive uncertainty estimation remains unchanged.
>
> For instance, in figure 3 and figure 5, SQR —
>
> - In figure 3 and figure 5, the PICP of all methods is 95%. Therefore, the PI generated from each method contains y (target) with equal probability. However, our method can cover y (target) with a much smaller width of PI. That implies our method is much more accurate than the other methods.
> - Note that high-quality PI should be as narrow as possible while containing some specified proportion ($\gamma$) of data points.
>
> The authors propose to use a mixture density network —
>
> - Disjoint $\gamma$% PI cannot be obtained as the closed-form from the Gaussian mixture, we proposed an auxiliary NN called the DDD network.
> - In the case of a single Gaussian distribution, \gamma% PI can be obtained in close form, $\mu$ $\pm$ $\alpha$ * $\sigma$ (e.g. $\alpha=1.96$ for $\gamma=0.95$). After obtaining $\gamma$% PI in closed form for all gaussian components, union can cover more than $\gamma$%, but often results in low-quality PIs.
> - For example, let's assume a GMM with two Gaussian components where the weight of one component is very small and the standard devation of the one is very large  $(\pi_1=0.99, \sigma_1=1, \pi_2=0.01, \sigma_2=100)$. In this case, it is very inefficient to unionize the same 95% interval from both components. The 95% intervals are $[\mu_1-1.96\sigma_1, \mu_1 +1.96\sigma_1]$, $[\mu_2-1.96\sigma_2, \mu_2 +1.96\sigma_2]$ and  so the width of PIs is 395.92. However, if you obtain an interval of 96% from a component with $\pi_1=0.99$, $[\mu_1 - 2.0537\sigma_1, + \mu_1+2.0537\sigma_1]$ becomes a 95% interval. In this case, the width of PI is 4.1074 which is much smaller than 395.92.
> - There is no closed-form solution to the problem of finding the $\gamma$% interval from GMM. Instead, our DDD method obtains a well-calibrated $\gamma$ % interval with the constrained optimization.
>
> We believe you overlooked the fact that the generation of high-quality disjoint PIs is a challenging task. Our method has enough novelty to do it effectively. We also believe that our study contributes to the field of predictive uncertainty estimation by demonstrating the issue of the inaccuracy of existing methods that have not yet been addressed.  This study can inspire much further research in the field of uncertainty estimation in deep learning.

---

> > ### Comment · Reviewer_3Zfh · 2021-11-23
> > **Response to authors**
> >
> > * Regarding evaluation and comparison to existing methods:
> >
> > I understand that generating disjoint PI's is a novel concept. However, my concern regarding evaluation still stands. There exist many methods that can produce multimodal predictive distributions. However, the proposed evaluation framework for disjoint PI's can only be applied to the proposed method. All other methods are evaluated as if they were unimodal. I believe that this entanglement between the proposed method and the evaluation makes it difficult to draw conclusions from the results.
> >
> > * Regarding the GMM:
> >
> > I believe the CDF of the GMM can be computed in closed form. If the GMM is well calibrated I would expect the PIs resulting from their inverse cfd to be somewhat reasonable. As a result, I think this could be a very simple baseline to prove that the proposed NN is helping. I understand the author's point that there could be a scenario where one of the components is very high variance and its PI engulfs that of the other component. However, would the proposed method deal with this any differently than the GMM iCDF? Perhaps that could be an interesting experiment to demonstrate the proposed approach.

---

> > > ### Author Response · Authors · 2021-11-27
> > > **Response to 3Zfh**
> > >
> > > **Regarding evaluation and comparison to existing methods:**
> > >
> > > I understand that generating disjoint PI's is a novel concept. However, my concern regarding evaluation still stands. There exist many methods that can produce multimodal predictive distributions. However, the proposed evaluation framework for disjoint PI's can only be applied to the proposed method. All other methods are evaluated as if they were unimodal. I believe that this entanglement between the proposed method and the evaluation makes it difficult to draw conclusions from the results.
> > >
> > > - Our argument is not that other methods assume unimodality, but that other methods produce single PI even in multimodal situations. As you commented, methods for approximating multimodal predictive distributions exist beyond mixture density networks. However, our goal is to accurately estimate predictive uncertainty by generating disjoint PIs. There is no method to generate disjoint PIs except for our method yet, so there was no other way than to compare it with single PI methods.
> > > - Extracting disjoint PIs from multimodal predictive distributions is a challenging problem for the following reasons. 1) We do not know the number of disjoint intervals for each data. 2) The loss function must be differentiable. We solved it effectively using the mixture density network and the DDD method.
> > > - The mixture density network can approximate the predictive distribution well enough if K is large enough. A mixture density network only needs to replace the last layer in the existing network. Therefore, our method can be used universally.
> > >
> > > **Regarding the GMM:**
> > >
> > > I believe the CDF of the GMM can be computed in closed form. If the GMM is well-calibrated I would expect the PIs resulting from their inverse CDF to be somewhat reasonable. As a result, I think this could be a very simple baseline to prove that the proposed NN is helping. I understand the author's point that there could be a scenario where one of the components is very high variance and its PI engulfs that of the other component. However, would the proposed method deal with this any differently than the GMM iCDF? Perhaps that could be an interesting experiment to demonstrate the proposed approach.
> > >
> > > - We use notation $F(x)$ as the CDF and $F^{-1}(x)$ as the inverse CDF below.
> > > - There is an effective way to compute the CDF of Gaussian, but it is not a closed-form expression because it contains integration [1]. Since CDF of GMM is a weighted sum of Gaussian, it is easy to calculate $F(x)$, but not a closed-form expression. Therefore, expressing the inverse CDF of GMM in closed form requires a numerical method. Section 3.2 of Salem et al., 2020 also states that it is numerically solved when finding the inverse CDF of a mixture of normal distribution.
> > > - Even assuming we know the $F^{-1}(x)$, generating $\gamma$ % disjoint PIs is a challenging task. This is because $F(x)$ are not symmetric and we need to produce multiple separate intervals which cover $\gamma$ %.
> > >
> > > We agree that it would be better to have a baseline method which deals with multimodality for performance comparison. However, recent studies have dealt with multimodal predictive distribution only with a single PI. We believe that the contribution of our method is sufficient just to show superiority compared to the state-of-the-art researches.
> > >
> > > Also, we initially tried to produce well-calibrated disjoint PIs in simple ways, but all failed. Generating disjoint PIs to estimate predictive uncertainty is a challenging task. We have solved it effectively with the DDD method.
> > >
> > >
> > > Reference
> > > [1] https://www.researchgate.net/profile/Naveen-Boiroju/publication/275885986_Approximations_to_Standard_Normal_Distribution_Function/links/554887b30cf2e2031b387faf/Approximations-to-Standard-Normal-Distribution-Function.pdf

---

### Official Review · Reviewer_RmVv · 2021-11-02

**Correctness:** 4
**Technical Novelty And Significance:** 2
**Empirical Novelty And Significance:** 2
**Recommendation:** 5
**Confidence:** 4

**Main Review:**

The strength of this paper is to construct better prediction intervals addressing the multimodality of predictive distribution and show better performance compared to the previous. It can be helpful to assess the uncertainty in a more rigid manner.

However, I have a doubt that the title ‘Distribution-driven disjoint uncertainty estimation for deep learning” is adequate. The paper uses deep architectures to generate the distribution-driven-disjoint PIs after the preprocessing of mixture density networks. The mixture of density networks is an architecture, and there is no solution or explanation when we use the other deep architectures for other tasks. This implies that the algorithm is not general to address the PIs.

In detail, I have some issues.

At first, I  feel that the figures in the paper are not kind to readers. In Figure 2, the prediction intervals are not shown well exception of the length of intervals, and in Figure 4, there is no explanation about solid curves after 0 on the x-axis.

Also, I have some issues with the proposed algorithms.
1) The use of $F_i$ seems reasonable. If you can provide more concrete evidence for the use of $F_i$, then the paper can be improved. More general criteria can be required for the generalization.
2) The $K$ in the mixture density network is a hyperparameter, and the empirical evidence for using $K=5$ is validated. Can you use the CV or other learning algorithm to choose the $K$?
3) I am interested in the computation time for Alg. 1. Is it efficient?
4) There is no detail concerning $M$ of ensembles.

Additionally, I cannot find the more significant use of disjoint PIs in various tasks. I was hoping you could provide more practical tasks related to PIs, such as active learning or calibration problems.


**Summary Of The Paper:**

This paper provides the algorithm for the construction of prediction intervals composed of disjoint intervals. The authors proposed the motivation of disjoint intervals well, and the motivating example is impressive. The algorithms are also well accommodated with the statistical or learning-based prediction intervals, which contribute to the assessment of the uncertainty of prediction in general.



**Summary Of The Review:**

The paper success in the construction of disjoint PIs. However, there is a lack of thorough insightful motivation, remain of some algorithmic issues. Presentation is not better.

---

> ### Author Response · Authors · 2021-11-15
> **Response to RmVv**
>
> [Response to main review]
>
> - Mixed density networks can be applied to general deep learning architectures simply by replacing the output nodes with Gaussian mixture nodes. The mean of the Gaussian mixture distribution works the same as the output node of the existing architecture.
> - Other PI-based predictive uncertainty estimation methods also replace the output node with a pair of a lower bound node and an upper bound node.
> - We agree that this needs to be added in the revised version to strengthen the motivation of our method.
>
> [Response to the Issues]
>
> At first, I feel that the figures —
>
> - The goal of Figure 2 is to show how the width of PIs (amount of uncertainty) of each model change in in-distribution and out-of-distribution. Shades will be expressed more clearly in the revised version.
> - The solid curve in Figure 4 represents a sample for which the approximated conditional density function \hat{p}(y|x) is multimodal. There are two modes.  PI with coverage probability provides that the left mode covers 43% and the right mode covers 54%. In the revised version, we will add more context about the sample to the caption.
>
> 1. The use of $F_i$  seems —
>     - The reason we approximate the predictive distribution $f_i$ as the GMM is that $F_i$ is not only differentiable but also generalizable. The reason is as follows.
>     - 1) Any distribution can be approximated with a Gaussian mixture model.
>     - 2) A mixture density network can be used in general by replacing the output nodes of a NN architecture with Gaussian mixture nodes.
>
> 2. The $K$ in the mixture density network is —
>     - When training a mixture density network, the objective of the training is to minimize negative log-likelihood (NLL). As $K$ increases, the NLL gradually decreases and then converges. For most of 11 UCI datasets, NLL converges when $K\leq5$.
>
> 3. I am interested in the computation time for Alg. 1. Is it efficient?
>     - Since our paper is the first to generate disjoint PIs and demonstrate the inaccuracy of the previous methods, we did not focus on time complexity.
>     - The computation time of algorithm 1 increases as K and M (=the number of models used for the ensemble) increase due to the union block. When we experimented with K=5 and M=5, it did not take long, so we did not consider the computation time.
>     - It is interesting to check how the computation time varies depending on the K and M. We will add it to the appendix of the revised version.
>
> 4. There is no detail concerning $M$  of ensembles.
>     - State-of-the-art papers (Lakshminarayanan et al., 2017, Salem et al., 2020) use $M=5$ without specific mention. We set $M=5$  for fair performance comparison with them.
>     - We will add how performance changes with ensemble number to the appendix of the revised version.
>
>
> Additionally, I cannot find —
>
> - PI is very effective in expressing the possible output range of y and measuring the amount of predictive uncertainty in the regression task. Our method can be applied in various domains mentioned in the first paragraph of the introduction section.
> - Our method can be applied by replacing the output nodes of the general NN architecture with Gaussian mixture nodes. Therefore, it is theoretically applicable to various tasks. However, further research is needed to deal with the uncertainty of multivariate output.
> - Since a single PI is used for predictive uncertainty estimation in several state-of-the-art papers, there is no need to question its practicality. In terms of practicality, there is no difference between disjoint PI and single PI.
>
> In summary, our method can be applied to various tasks to effectively measure predictive uncertainty.
>
> We believe that this study can inspire much further research in the field of uncertainty estimation in deep learning. Please consider the contribution and novelty of this paper under the premise that the presentation will be improved in the revised version.
>
> Our contribution and novelty are as follows. 1) We pointed out for the first time that state-of-the-art methods can be inaccurate to use a single prediction interval for predictive uncertainty estimation. 2) We propose a method for generating disjoint prediction intervals for the first time. 3) Our method enables much more accurate uncertainty estimation than existing methods.
>
> We appreciate your insightful feedback on ways to strengthen our paper. Thanks to your feedback, we can further emphasize the motivation and contribution of our method in the revised version.

---

> > ### Comment · Reviewer_RmVv · 2021-11-22
> > **Reply to Rebuttal**
> >
> > Thanks for your detailed reply. My concern is that the PI for deep networks. I'm not sure that we can obtain ${\it reliable}$ PIs by the use of mixture density networks in the output nodes.  If the aim is only optimized PIs, then I am not sure that the tern of `PIs for deep learning' in the title is appropriate. Maybe 'optimized PIs by deep networks' can be appropriate.

---

> > > ### Author Response · Authors · 2021-11-24
> > > **Thanks for the good comments**
> > >
> > > Thanks for the good comments. We agree that it would be more appropriate to rename it 'Distribution-Driven Disjoint Prediction Intervals for Predictive Uncertainty Estimation'.

---

### Official Review · Reviewer_8k1c · 2021-11-02

**Correctness:** 3
**Technical Novelty And Significance:** 2
**Empirical Novelty And Significance:** 2
**Recommendation:** 5
**Confidence:** 2

**Main Review:**


Strengths:
- Studies an interesting and well-motivated problem.
- Contains thorough quantitative and qualitative experiments to both motivate the problem and show improvement over the prior work.

Weaknesses:
[Presentation]
- The writing needs a lot of improvements. Many sentences do not follow the scientific writing style. Here I only mention the typos but there are many other cases where the sentences need to be re-written in a more elegant way.
- Fourth line of Abstract: it -> they
- Third line of introduction: something missing in the sentence probably you meant: functions "+of" the NN
- Last paragraph of page 1: a single continuous PI provides low-quality PIs -> a single continuous PI "+method?" provides low-quality PIs. In many other cases, PI methods and PIs are used interchangeably which should not be the case.
- Last sentence in section 2: The dot before (so-called MVE_ens) should be omitted.
- First paragraph of page 4: which reflects HQ principle and "+is" differentiable.
- After eq.6: Then, the coverage probability... -> why "then"?
- After eq.6: L_PICP is defined as the mean square error with the target proportion γ with the max(0, x) operation. -> some words missing?
- Before eq.7: number of the mixture -> number of the "components"
- Last paragraph of 4.2: λ is a hyperparameter that determines how much trade-off between L_PICP and L_MPIW. -> some words missing?
- Last paragraph of 4.2: If PICP obtained through training is less than γ, increase λ -> writing should be improved.
- First sentence of 4.3 needs to be re-written
- In the Hyperparameters section: All neural networks... -> all neural networks
- last two paragraphs of 5.2 need to be re-written
- title of 4.2: typo in architecture

[Questions]
- In Figure 2, why is the blue line sometimes showing a negative value while it should correspond to the width of the prediction interval?
- In Figure 2, how do you justify your choice of training and testing samples? You keep the points with x>4 or x<-4 but should it not be the case that you shuffle your points and randomly pick a percentage as test data points? The regressor is never trained on points with x>4 or x<-4 while asked to predict them.
- Figure2 is missing legend.
- Figures 3,4 are missing axes labels and ticks.
- Why are the experiments for the Protein dataset repeated 5 times and the other datasets repeated 20 times?


[Other Concerns]
- You mention you normalize the target distribution based on the entire data but you have to rather normalize based on only the training data.
- There is not enough details to fully understand your method. I do not find the exact mathematical definition of F_i which is an important part of your approach.
- A discussion regarding the time complexity of the algorithm in terms of the number of components in the mixture model (K) is missing.

**Summary Of The Paper:**

The paper addresses the problem of determining prediction intervals (PI) in regression task. The prediction interval problem can be summarized as predicting a lower and upper bound between which the potential observation falls with a certain probability. The paper proposes a method to report the prediction interval as the union of disjoint intervals, in contrast with the previous methods which report a unified continuous interval. The motivation is that if the conditional density function has multiple modes, a single prediction interval may not be well descriptive of the uncertainty of the predictive model. To achieve this goal, they propose a differentiable objective function together with a Neural Network architecture that produces the union of disjoint prediction intervals. Through experiments, they show that multimodality often exists in real-world datasets and that their method manages to produce prediction intervals of higher quality (in terms of the commonly used metrics to assess the quality of the prediction intervals such as coverage probability and interval width) compared to the previous work.

**Summary Of The Review:**

While the paper studies an interesting problem, it suffers from presentation issues which makes it hard to grasp the main technical contribution of the paper. The paper seems to be written in a rushed way and consists of many sentences which have missing words or are grammatically incorrect. This gets bolder as we move on to the later parts of the paper. I tried to mention some of the typos and the incomplete sentences as I went through the paper but they are too many to mention in this review. Many parts of the paper need to be re-written in order for it to get ready for publication.

The novelty is in re-writing the objective function, which tries to minimize prediction interval length while maintaining a certain coverage probability, in a differentiable manner and then optimizing it.  Some technical details are also missing (such as discussion on the complexity of the approach). Overall I think the paper is not ready for publication and needs improvements both in technical content and presentation.

---

> ### Author Response · Authors · 2021-11-15
> **Response to Reviewer 8k1c**
>
> Thank you for your insightful feedback on ways to strengthen our paper.
>
> We tried to explain the details in the captions because it is ambiguous how to display the axes, labels, and tick marks when multiple prediction intervals and conditional densities are included in one figure. However, it seems that our explanation was not enough, as another reviewer RmVv also pointed out that our figure was not friendly. We will correct the issues about grammar and figures you pointed out in the revised version.
>
> Our contribution and novelty are as follows. 1) We pointed out for the first time that state-of-the-art methods can be inaccurate to use a single prediction interval for predictive uncertainty estimation. 2) We propose a method for generating disjoint prediction intervals for the first time. 3) Our method enables much more accurate uncertainty estimation than existing methods.
>
> We believe that this study can inspire much further research in the field of uncertainty estimation in deep learning. Please consider the contribution and novelty of this paper under the premise that the presentation will be improved in the revised version.
>
> [Reponse to Questions]
>
> In Figure 2, why is the blue line sometimes showing a negative value while it should correspond to the width of the prediction interval?
>
> - The right axis represents the width of the prediction interval (blue line) and the left axis represents the y value of the data point. As you said the width of the prediction interval should always be positive, the right axis starts at zero. To avoid confusion, we will add a description of this to the caption.
>
> In Figure 2, how do you justify your choice of training and testing samples? You keep the points with x>4 or x<-4 but should it not be the case that you shuffle your points and randomly pick a percentage as test data points? The regressor is never trained on points with x>4 or x<-4 while asked to predict them.
>
> - This experiment was set up very similar to Lakshminarayanan et al., 2017 and Pearce et al., 2018. It is an experiment to show that the ensemble model predicts predictive uncertainty better than single models in out-of-distribution. A desirable predictive uncertainty estimation model should predict an increasingly larger amount of uncertainty as it moves away from the in-distribution. To understand this intuitively, we used one-dimensional toy data, and the boundary between in-distribution and out-of-distribution is clearly defined (x=4, x=-4).
> - No test data points are needed in this experiment (we only calculate the width of the prediction interval from the out-of-distribution). When setting $\lambda$ so that PICP satisfies $0.95 \pm 0.01$, only training data is used.
>
> Figure2 is missing legend. Figures 3,4 are missing axes labels and ticks.
>
> - Comments on the figure will be reflected in the revised version.
>
> Why are the experiments for the Protein dataset repeated 5 times and the other datasets repeated 20 times?
>
> - We used the experimental conditions of the SNM-QD+ paper as it is in order to bring the experimental results of SNM-QD+ as it is (due to the difficulty of hyperparameter tuning of SNM-QD+). SNM-QD+ repeated the protein dataset only 5 times.
> - The protein dataset contains 45730 samples which is much more than other datasets so that it takes a longer time for the training process than others.
>
> [Response to Concerns]
>
> You mention you normalize the target distribution based on the entire data but you have to rather normalize based on only the training data.
>
> - Input normalization for performance improvement was normalized based on the training set.
> - The normalization of target y based on the entire dataset is to standardize the result. Standardization shows how large the width of PI is compared to the standard deviation of target y. Pearce et al., 2018;, Salem et al., 2020 also standardize target y on the entire dataset.
>
> There is not enough details to fully understand your method. I do not find the exact mathematical definition of F_i which is an important part of your approach.
>
> - $F_i$ is the cumulative density function of the Gaussian mixture distribution in (7). It is the weighted sum of the cumulative density function of the Gaussian distribution.
>
> A discussion regarding the time complexity of the algorithm in terms of the number of components in the mixture model (K) is missing.
>
> - Since our paper is the first paper to propose a method for estimating predictive uncertainty by generating disjoint PIs and to demonstrate the inaccuracy of the previous methods, we did not focus on time complexity.
> - The computation time of algorithm 1 increases as K and M (=the number of models used for the ensemble) increase due to the union block. When we experimented with K=5 and M=5, it didn't take long, so we didn't consider the computation time.
> - It is interesting to check how the computation time varies depending on the K and M you proposed. We will add it to the appendix of the revised version.

---

### Decision · Program_Chairs · 2022-01-20

**Decision:**

Reject

**Comment:**

This paper presents a method for producing a mixture of (disjoint) predictive distributions for deep learning models rather than a single predictive distribution.  The reviewers in general found that the idea had strong potential, was well motivated and addresses an important and under-appreciated problem in deep learning.  They seemed to find the proposed approach of using mixture density networks to be sensible.  However, the reviewers seemed to find that the paper was unclear in presentation and grammatically, as if hastily written.  One reviewer noted that they would not be able to reproduce the method given the confusing presentation.  The reviewers also found that the experiments didn't adequately evaluate their method empirically.  Unfortunately, the reviewers all agreed that the paper is not quite ready for publication (5, 3, 5).  Careful rewriting of the paper and the technical contributions and strengthening the experiments would go a long way towards improving this paper for a future submission.